# On the Adversarial Robustness of Discrete Image Tokenizers

## Abstract

Discrete image tokenizers encode visual inputs in a sequence of tokens from a finite vocabulary. Pre-trained tokenizers, typically trained together with a decoder for image reconstruction, are an increasingly popular alternative to CLIP image encoders for multimodal systems, including encoder-only, encoder-decoder and decoder-only models. However, unlike CLIP encoders, their vulnerability to adversarial attacks has not been explored. Ours being the first work studying this topic, we first formulate attacks that aim to perturb the features extracted by discrete tokenizers, and thus change the extracted tokens. Since the attacks target only the image encoding, they are computationally efficient, agnostic of the downstream application, and effective on classification, multimodal retrieval and captioning tasks. Second, to defend against this vulnerability, inspired by recent work on robust CLIP encoders, we fine-tune popular tokenizers with unsupervised adversarial training, while keeping all other components frozen. While unsupervised and task-agnostic, our approach significantly improves robustness to both unsupervised and end-to-end attacks. Unlike standard supervised adversarial training, our method can generalize well to unseen tasks and data while also being able to directly leverage any amount of unlabeled images. Overall, our work demonstrates that the resistance of image tokenizers to adversarial attacks strongly impacts the robustness in downstream tasks, and presents an important step in developing generalizable and safe multimodal foundation models.

## 1 Introduction

Similar to text tokenizers, discrete image tokenizers represent visual input as a sequence, typically of fixed length, of vectors from a finite codebook (Van Den Oord et al., 2017; Yu et al., 2024b; Ma et al., 2025; Bachmann et al., 2025). However, unlike text tokenizers such as BPE (Gage, 1994), image tokenizers rely on deep networks, which are trained, together with a de-tokenization model, for image reconstruction. These de-tokenizers can then be used for synthetic image generation in autoregressive frameworks (Ma et al., 2025; Xie et al., 2025; Wang et al., 2024), which are competitive with diffusion models. Discrete image tokenizers have recently become part of more complex architectures as image encoders, representing an increasingly popular alternative to CLIP (Radford et al., 2021) or DINO (Oquab et al., 2023) image encoders. Multimodal encoder-decoder models such as Unified-IO (Lu et al., 2022), 4M (Mizrahi et al., 2023) and 4M-21 (Bachmann et al., 2024) employ discrete tokenizers for both natural images and segmentation maps, while FuseLIP (Schlarmann et al., 2025) leverages them to enable a multimodal embedding model, based on early-fusion of text and image inputs, trained with a contrastive loss. These models can be applied, either zero-shot or via transfer learning, on a variety of unimodal and multimodal tasks, including classification, retrieval, and VQA. Moreover, discrete image tokenizers are used by decoder-only generative models (Chameleon Team, 2024; Xie et al., 2025; Wang et al., 2025; Ma et al., 2025). In fact, the finite vocabulary enables language-style autoregressive modeling, yielding unified understanding and generation in both vision and language domains. As a consequence, the robustness (or lack of it) of image tokenizers to adversarial attacks directly influences how vulnerable all such models are. However, while several works have focused on testing and improving the adversarial robustness of standard, (e.g., CLIP), encoders, this aspect of discrete image tokenizers remain completely unexplored.

In this work, we first study the vulnerability of recently proposed discrete image tokenizers, both in isolation and as part of larger models. In particular, we test *unsupervised attacks* which aim to alter

the tokenization of the original image by distorting the encoding and consequently assigning wrong codebook vectors. Since these attacks target only the image tokenizer, the resulting perturbations are agnostic to the downstream tasks for which the tokens are used (e.g., reconstruction, classification, language generation). We show that such direct unsupervised attacks are often effective against standard tokenizers, and in some cases even competitive with end-to-end supervised attacks, which require task-specific information (e.g., labels) and are more computationally expensive because they target the entire system. For instance, in autoregressive multimodal models, the large language model (LLM) has one to two orders of magnitude more parameters than the image tokenizer. Notably, our unsupervised attacks can make an LLM output a target (malicious) caption for a given input image by matching the tokenization of the input image to that of a target image semantically aligned with the desired caption, without requiring direct access to the LLM itself. These results highlight the vulnerability of existing discrete image tokenizers, and the safety risks they pose to downstream models across a variety of tasks.

Then, in order to mitigate this vulnerability, we extend the framework of Schlarmann et al. (2024), introduced for robust CLIP vision encoders, to image tokenizers. In particular, we fine-tune discrete image tokenizers with unsupervised adversarial training, i.e., we train the model to yield consistent tokenization for the original images and their corresponding adversarial counterparts, computed on-the-fly by unsupervised attacks. Since this approach is independent of any downstream task, the resulting tokenizer can be directly plugged backed in any system which relied on the original tokenizer without the need of additional adaptations. In fact, through extensive experiments, we show that unsupervised adversarial fine-tuning increases the robustness to end-to-end attacks in a variety of tasks, from image classification to captioning. In particular, by fine-tuning TiTok tokenizers, we obtain robust FuseLIP models for multimodal embedding, while with adversarial training of UniTok we get a multimodal LLM (MLLM) robust on VQA and captioning tasks. Finally, we provide additional fine-grained comparisons: we analyze the effect of the unsupervised attacks by reconstructing the adversarial images, which reveals quite significant differences across tokenizers, the differences between supervised and unsupervised adversarial training, and the role of the training dataset for adversarial fine-tuning.

**Contributions.** In summary, (a) our work is the first to systematically test and improve the adversarial robustness of discrete image tokenizers. (b) To evaluate their vulnerability, we propose unsupervised attacks that are both efficient and task-agnostic. (c) Importantly, we also show that the same attacks can be leveraged to adversarially fine-tune these tokenizers. (d) Our unsupervised adversarial fine-tuning robustifies the tokenizers against both unsupervised and end-to-end supervised attacks, at a significantly lower computational cost than end-to-end supervised adversarial fine-tuning. (e) In contrast to supervised defenses, our defense can directly leverage any amount of unlabeled data. (f) These robust tokenizers can be seamlessly integrated as image encoders in larger systems, strengthening the robustness of multimodal embedding models (FuseLIP) and MLLMs (UniTok-MLLM) to all attacks (supervised or not) across diverse tasks well beyond the training data. Therefore, our work identifies the crucial role of tokenizers in safeguarding multimodal models while offering a practical step toward building more robust and safe multimodal foundation models.

## 2 RELATED WORK

**Image tokenizers.** Image tokenizers aim to compress visual data via deep learning models, trained with image reconstruction objectives possibly aided by auxiliary losses. Continuous tokenizers encode images into features embedded in a continuous space (Fan et al., 2025). Conversely, discrete tokenizers convert images into sequences of tokens drawn from a fixed vocabulary, typically via vector quantization. VQ-VAE (Van Den Oord et al., 2017) introduced a vector-quantized bottleneck, replacing continuous latents with discrete categorical codes. Subsequent works enhanced reconstruction quality through adversarial training (VQGAN (Yu et al., 2022)), transformer architectures (ViT-VQGAN (Yu et al., 2022), Efficient-VQGAN (Cao et al., 2023)), multi-stage quantization (RQ-VAE (Yu et al., 2022), MoVQ (Zheng et al., 2022)), and lookup-free schemes (MAGVIT-v2 (Yu et al., 2024a), FSQ (Mentzer et al., 2024)) for improved efficiency and expressiveness. While early methods encode images into 2D grids of latents, recent works have developed 1D tokenizers which return a sequence of tokens without any spatial structure (Yu et al., 2024b; Miwa et al., 2025; Bachmann et al., 2025; Ma et al., 2025). Among those, TiTok (Yu et al., 2024b) demonstrates that as few as 32 tokens can suffice for high-quality reconstructions. Recently, UniTok (Ma et al., 2025) focuses on building representations effective for both generation and understanding by integrating

reconstruction and CLIP supervision during training. UniTok introduces multi-codebook quantization and attention-based projection, enhancing latent space expressivity and enabling tokenizers to produce representations that generalize well across both generative and discriminative tasks. While TiTok and UniTok tokenizers have a fixed number of tokens, other approaches such as One-D-Piece (Miwa et al., 2025) and FlexTok (Bachmann et al., 2025) allow varying the number of tokens to trade-off compression rate and reconstruction quality at inference time.

**Discrete image tokenizers in downstream tasks.** While their corresponding de-tokenizers are commonly used for image generation in frameworks such as MaskGIT (Chang et al., 2022), discrete image tokenizers have been recently used as components of models for downstream tasks. First, the features extracted by the tokenizers can be used for transfer learning in classification tasks, either zero-shot (UniTok (Ma et al., 2025)) or with linear probing (FlexTok (Bachmann et al., 2025)). Second, encoder-only (FuseLIP (Schlarmann et al., 2025)) and encoder-decoder models (Mizrahi et al., 2023; Bachmann et al., 2024; Lu et al., 2022) leverage discrete image tokenizers to extend masked modeling losses to visual inputs, beyond language data. Finally, multiple early-fusion autoregressive models, such as Chameleon (Chameleon Team, 2024), Show-o (Xie et al., 2025), UniTok-MLLM (Ma et al., 2025), SelfTok (Wang et al., 2025) use discrete tokenizers to convert input images into a sequence of tokens from a fixed codebook, providing a setup similar to natural language.

**Adversarial robustness of image encoders.** Adversarial robustness to $\ell_p$-bounded attacks has been extensively studied for a variety of vision tasks, such as image classification (Croce et al., 2020), semantic segmentation (Croce et al., 2024), object detection (Li et al., 2025), with the development of many algorithms to both generate adversarial perturbations (Carlini & Wagner, 2017; Croce & Hein, 2020) as well as improve the robustness to deep learning models (Madry et al., 2018; Zhang et al., 2019). With the development of foundation multimodal models such as CLIP (Radford et al., 2021) and SAM (Kirillov et al., 2023), those approaches have been adapted to modern scenarios. Moreover, new techniques have been designed to attack multimodal autoregressive large language models such as LLaVA (Li et al., 2024) by perturbing the input images, which allows an attacker to control the output generated by the model (Schlarmann & Hein, 2023; Qi et al., 2024). These works reveal the persisting vulnerability of image encoders to adversarial perturbations, both in isolation and when used as part of more complex systems (Bhagwatkar et al., 2024). Therefore, a few works have proposed methods to improve their robustness, typically leveraging different forms of adversarial training (Madry et al., 2018). For example, Mao et al. (2023); Schlarmann et al. (2024) fine-tuned the vision encoder of CLIP with variants of adversarial training. However, to our knowledge, the robustness of (discrete) image tokenizers has not been studied so far.

## 3 Testing and Improving the Adversarial Robustness of Image Tokenizers via Unsupervised Attacks

In the following, we first provided a short background on discrete image tokenizers, then introduce our unsupervised attacks as well as experiments supporting their effectiveness. Finally, we present how these can be leveraged to improve the robustness of tokenizers.

**Background.** Discrete image tokenizers extract a sequence of $T$ $d$-dimensional latent vectors from an input image. Concretely, they first leverage an encoder $\phi : I \to \mathbb{R}^{T \times d}$, typically a CNN or vision transformer, that maps an image $x$ to $T$ embeddings, $\phi(x) = \{h_i\}_{i=1}^T \in \mathbb{R}^{T \times d}$. Then, a vector quantizer with a learned codebook $\mathcal{C} = \{e_k \in \mathbb{R}^d\}_{k=1}^K$ replaces each pre-quantization embedding $h_i \in \mathbb{R}^d$ (at location $i$) with its nearest code, i.e., $q_i = \arg\min_{k \in [K]} \|h_i - e_k\|_2$. This yields a discrete index map $\{q_i\}_{i=1}^T$. Unlike continuous encoders that output continuous space embeddings and remain differentiable everywhere, nearest-neighbor quantization is piecewise constant since all embeddings with a particular nearest code are mapped to it, creating discretized cells in the continuous latent space. Hence, even infinitesimal shifts at the boundaries of these cells can lead to a new nearest code and index. In practice, the encoder and codebook are learned jointly using straight-through updates.

### 3.1 Unsupervised attacks via embedding distortion

Pre-trained image tokenizers are used as plug-ins in complex systems for multiple downstream tasks. Therefore, we aim to develop attacks that can be effective regardless of the downstream application.

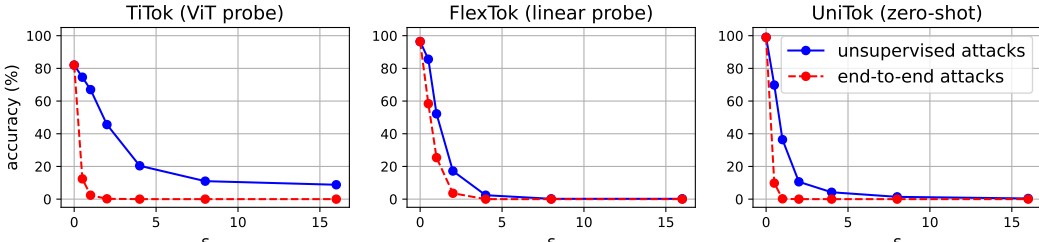

Figure 1: **Unsupervised vs supervised adversarial attacks for classification.** We report the robust accuracy when varying the perturbation radius $\epsilon$ (scaled to [0, 255]) for three classifiers on Imagenette: TiTok with ViT probing (left), FlexTok with linear probing (middle) and zero-shot UniTok (right). In most cases, our unsupervised attacks (blue curves), which target only the image tokenizer and do not need label information, perform close to end-to-end attacks (red) which target the entire classifier and need label information. For small $\epsilon$ values, our unsupervised attacks are slightly worse than the supervised attacks. Both attacks are optimized with 100 iterations of APGD on 500 images.

Since the representation of an image $x$ used by any downstream model is determined by the codebook tokens output by the tokenizer, we expect that changing the tokens will lead to an uninformative or distorted encoding of $x$. While we could aim to directly change the index sequence returned by the tokenizer, the indices themselves carry no (or minimal) semantic or perceptual information, and the resulting problem is not differentiable. Hence, we propose an attack in the pre-quantization embedding space of discrete image tokenizers. Specifically, we maximize the $\ell_2$-distance between the embeddings of the clean and perturbed images, similar to existing attacks on continuous encoders (Croce & Hein, 2024; Schlarmann et al., 2024), i.e., we aim to solve

$$\max_{\|\delta\|_p \leq \epsilon} \sum_{i=1}^{T} \|h_i(x + \delta) - h_i(x)\|_2^2, \tag{1}$$

where $h_i(\cdot) \in \mathbb{R}^d$ is the $i$-th pre-quantization embedding produced by the image encoder, and $\delta$ is the adversarial perturbation constrained by an $\ell_p$-norm bound $\epsilon$. Intuitively, our approach aims to shift the encoder output sufficiently far in latent space to alter the resulting quantized codes, thereby corrupting any downstream application of the tokens. We remark that our objective directly manipulates the encoder output (prior to vector quantization), without requiring access to the codebook or any information (e.g., class labels) of the downstream applications.

**Effectiveness of unsupervised attacks.** To assess the effectiveness of our unsupervised attack, we first compare it to end-to-end attacks on image classification. We obtain three, diverse classifiers based on discrete tokenizers on the Imagenette dataset (Howard, 2019): we train a ViT probe on the quantized features of TiTok-BL128 (Yu et al., 2024b), a linear probe on FlexTok (Bachmann et al., 2025), and zero-shot UniTok (Ma et al., 2025). We compare unsupervised attacks, where APGD (Croce & Hein, 2020) is used to optimize Eq. (1), to a standard end-to-end attack (APGD on the cross entropy loss) which targets the entire classifier and requires label information, both with 100 iterations. Fig. 1 shows the robust accuracy of the classifiers when varying the radius $\epsilon$ of the $\ell_\infty$-bounded attacks. Notably, our unsupervised attacks (blue curves) often perform close to the end-to-end attacks (red), lagging behind particularly for small $\epsilon$ values. This also shows that even at limited computational cost (only 100 iterations), unsupervised attacks are effective in fooling the classifiers, and hence are useful for adversarial training (see Sec. 3). Finally, we see that the classifiers built on discrete tokenizers are highly vulnerable to adversarial perturbations.

**Reconstruction of adversarial images.** A unique aspect of our setup is that we can use the image de-tokenizers to reconstruct the adversarial images derived by unsupervised attacks. We qualitatively estimate how distorting the features extracted by the tokenizer's encoder impacts the reconstruction. In Fig. 2 we provide several examples of reconstruction of unsupervised attacks (2500 iterations of APGD) at different $\epsilon$. Despite these attacks being similarly effective on classification (as mentioned above), they affect reconstruction differently depending on the tokenizer, with TiTok yielding the most distorted reconstructed images while FlexTok being the most robust with still clearly recognizable subjects. This hints to some structural difference among image tokenization approaches which might impact robustness, presenting an interesting direction for future work.

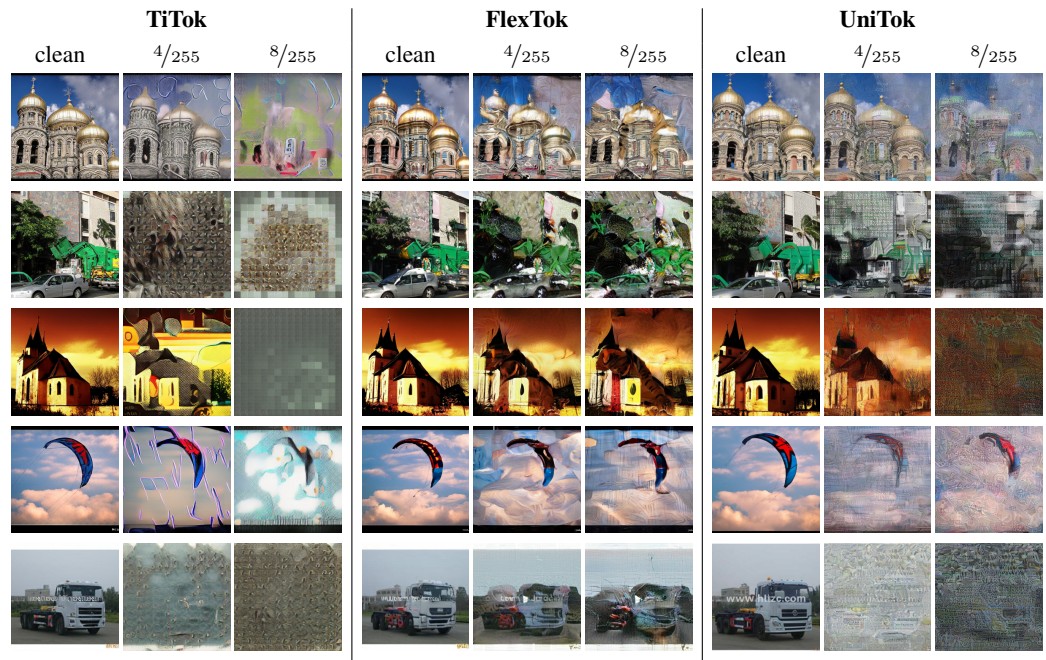

Figure 2: **Reconstruction of unsupervised attacks.** For each tokenizer, we show the reconstruction (given by the corresponding de-tokization models) of the clean images and adversarial images computed by unsupervised attacks at $\epsilon = {}^4/255, {}^8/255$ with 2500 steps of APGD. The perturbed inputs affect the reconstruction differently depending on the tokenizer, with TiTok yielding the most distorted decoded images while FlexTok being most robust, with still clearly recognizable subjects.

## 3.2 ROBUST TOKENIZERS VIA UNSUPERVISED ADVERSARIAL FINE-TUNING

A common approach to improve the robustness of neural networks is adversarial training (Madry et al., 2018), where adversarial perturbations are generated on the fly during training and used to augment the original training set. In our case, we want to make the image tokenization invariant to adversarial perturbations while preserving its effectiveness for downstream applications. Inspired by the approach of Schlarmann et al. (2024) for improving the robustness of CLIP's image encoder, we propose adversarially fine-tuning the image tokenizer on our unsupervised attack. This yields

$$\min_{\theta} \frac{1}{|\mathcal{D}|} \sum_{x \in \mathcal{D}} \max_{\|\delta\|_p \leq \epsilon} \sum_{i=1}^{T} \left\| h_i^{\theta}(x + \delta) - h_i^{\theta_{\mathrm{orig}}}(x) \right\|_2^2, \tag{2}$$

as the training objective, where $\theta$ are the parameters of the fine-tuned tokenizer, $\theta_{\mathrm{orig}}$ the parameters of the original tokenizer (before fine-tuning), and $\mathcal{D}$ is the training dataset. This loss pushes the fine-tuned tokenizer to yield consistent embeddings in an $\ell_p$-ball of radius $\epsilon$ around the embeddings given by the original tokenizer. Therefore, the fine-tuned tokenizers could readily replace the original tokenizers in downstream tasks without hurting clean performance while improving robustness. In principle, the quantization steps of discrete tokenizers make it possible to preserve the downstream performance even without exactly solving Eq. (2), since sufficiently close embeddings lead to the same codebook indices. Compared to task-specific end-to-end adversarial training, our task-agnostic approach has several advantages: *(i)* it yields tokenizers which can be deployed across multiple downstream applications, *(ii)* can leverage virtually any image dataset, even beyond that used for pre-training (as we show in Sec. 5) since it does not need label information, and *(iii)* fine-tunes only a subset of the parameters, i.e., those of the tokenizer's encoder, substantially lowering the computational cost.

## 4 EXPERIMENTS

In the following, we present results for the original and our adversarially fine-tuned tokenizers across tasks and datasets. In detail, we experiment with two vector-quantized image tokenizers, TiTok-

Table 1: **Evaluation of FuseLIP on image classification and multimodal retrieval.** We report the clean and robust accuracy (%) under $\ell_\infty$-bounded perturbations with $\epsilon \in \{2/255, 4/255\}$ for original and robust tokenizers trained on different radii. The rightmost block reports averages across datasets.

| Tokenizer | Imagenette | | | Caltech101 | | | OI-Crop | | | OI-Pos | | | Average | | |
|---|---|---|---|---|---|---|---|---|---|---|---|---|---|---|---|
| | clean | $\ell_\infty$ | | clean | $\ell_\infty$ | | clean | $\ell_\infty$ | | clean | $\ell_\infty$ | | clean | $\ell_\infty$ | |
| | | $2/255$ | $4/255$ | | $2/255$ | $4/255$ | | $2/255$ | $4/255$ | | $2/255$ | $4/255$ | | $2/255$ | $4/255$ |
| clean | 93.6 | 2.6 | 0.0 | 74.4 | 0.6 | 0.0 | 71.8 | 7.4 | 0.8 | 69.2 | 5.4 | 1.4 | 77.3 | 4.0 | 0.6 |
| AT$^{4/255}$ | 92.2 | 63.2 | 36.0 | 73.0 | 48.2 | 19.8 | 65.4 | 49.4 | 26.6 | 67.2 | 47.6 | 23.8 | 74.5 | 52.1 | 26.6 |
| AT$^{8/255}$ | 89.0 | 67.8 | 46.4 | 73.2 | 48.6 | 32.4 | 59.6 | 47.8 | 35.0 | 65.2 | 53.6 | 34.4 | 71.8 | 54.5 | 37.1 |
| AT$^{12/255}$ | 86.4 | 68.4 | 50.4 | 67.6 | 47.6 | 36.0 | 54.4 | 49.0 | 37.2 | 62.6 | 49.4 | 34.8 | 67.8 | 53.6 | 39.6 |
| AT$^{16/255}$ | 81.6 | 62.2 | 47.8 | 59.6 | 46.2 | 35.4 | 50.0 | 46.2 | 33.4 | 56.8 | 47.6 | 37.4 | 62.0 | 50.6 | 38.5 |

Table 2: **Evaluation of UniTok on image classification.** For each dataset, we report the accuracy (%) under no attack (clean) and $\ell_\infty$-bounded perturbations with $\epsilon \in \{2/255, 4/255\}$ for original and robust tokenizers trained on different radii. The rightmost block reports averages across datasets.

| Tokenizer | Imagenette | | | Caltech101 | | | ImageNet | | | Average | | |
|---|---|---|---|---|---|---|---|---|---|---|---|---|
| | clean | $\ell_\infty$ | | clean | $\ell_\infty$ | | clean | $\ell_\infty$ | | clean | $\ell_\infty$ | |
| | | $2/255$ | $4/255$ | | $2/255$ | $4/255$ | | $2/255$ | $4/255$ | | $2/255$ | $4/255$ |
| clean | 99.2 | 0.0 | 0.0 | 85.7 | 0.0 | 0.0 | 67.3 | 0.0 | 0.0 | 84.1 | 0.0 | 0.0 |
| AT$^{4/255}$ | 99.0 | 91.3 | 75.2 | 81.3 | 51.8 | 23.2 | 66.5 | 33.5 | 14.1 | 82.3 | 58.9 | 37.5 |
| AT$^{8/255}$ | 97.4 | 92.1 | 81.9 | 77.8 | 58.5 | 42.9 | 58.5 | 37.5 | 24.6 | 77.9 | 62.7 | 49.8 |
| AT$^{12/255}$ | 95.6 | 91.1 | 82.3 | 72.2 | 56.7 | 46.6 | 50.2 | 35.9 | 25.0 | 72.7 | 61.2 | 51.3 |
| AT$^{16/255}$ | 91.5 | 87.5 | 79.2 | 65.1 | 54.2 | 43.8 | 41.7 | 30.6 | 23.0 | 66.1 | 57.4 | 48.7 |

BL128 and UniTok, since both, unlike FlexTok, are part of larger models and can be tested on multiple downstream tasks. In addition, we report the results of the smaller TiTok-S128 in App. B. To obtain the robust versions, we adversarially fine-tune only the encoder of each tokenizer, while keeping the codebook, downstream decoders, LLMs or any other components frozen. We fine-tune for one epoch under $\ell_\infty$-bounded perturbations with our unsupervised embedding-space attack (50 steps of APGD), see Sec. 3, with perturbation radii $\epsilon \in \{4/255, 8/255, 12/255, 16/255\}$, on ImageNet-1k. More details on the setup are in App. A. Remarkably, despite adversarially fine-tuning only on ImageNet-1k, the resulting robust tokenizers exhibit strong robustness across a wide range of downstream datasets and tasks, highlighting their ability to transfer robustness beyond the training domain.

### 4.1 ROBUST TOKENIZERS LEAD TO ROBUST EMBEDDING MODELS

First, we test the effect of replacing the original tokenizer's encoder with our adversarially fine-tuned version in embedding models, for classification and multimodal retrieval tasks.

**FuseLIP.** Schlarmann et al. (2025) build FuseLIP, a family of early-fusion multimodal embedding models obtained via contrastive learning, relying on TiTok tokenizers as frozen image encoders. Thanks to this framework, we can evaluate the robustness of FuseLIP with the original and our adversarially fine-tuned TiTok encoders on a variety of zero-shot downstream tasks. In Table 1, we report the results of FuseLIP on two image classification datasets, Imagenette and Caltech101 (Fei-Fei et al., 2004), and two multimodal retrieval datasets (OI-Crop and OI-Pos) (Schlarmann et al., 2025), which require encoding image-text pairs in a single embedding vector. We evaluate robustness with end-to-end attacks at $\epsilon = 2/255$ and $4/255$. For classification, we use the popular AutoAttack (Croce & Hein, 2020), while for multimodal tasks we use 100 iterations of APGD on the cross-entropy loss. In both cases, we use a straight-through estimator to bypass the non-differentiable quantization step. In all datasets, we observe that zero-shot models, for either classification or retrieval, built on the original tokenizers exhibit no robustness to adversarial attacks. In contrast, our robust tokenizers, in any configuration, significantly improve adversarial robustness across all datasets. In particular, the training radius provides explicit control over the robustness–accuracy trade-off: tokenizers trained at lower radii ($\epsilon = 4/255$) retain clean accuracy much closer to the original models, while those trained

Table 3: **UniTok-MLLM on VQA.** We report the accuracy (%) on VQAv2, OK-VQA, and GQA under no attack (clean) and $\ell_\infty$-bounded perturbations with $\epsilon \in \{2/255, 4/255\}$ for original and robust tokenizers trained on different radii. The rightmost block reports averages across datasets.

| Tokenizer | VQAv2 | | | OK-VQA | | | GQA | | | Average | | |
|---|---|---|---|---|---|---|---|---|---|---|---|---|
| | clean | $\ell_\infty$ | | clean | $\ell_\infty$ | | clean | $\ell_\infty$ | | clean | $\ell_\infty$ | |
| | | $2/255$ | $4/255$ | | $2/255$ | $4/255$ | | $2/255$ | $4/255$ | | $2/255$ | $4/255$ |
| clean | 73.2 | 14.4 | 8.5 | 59.6 | 2.1 | 0.8 | 68.0 | 13.4 | 10.0 | 66.9 | 10.0 | 6.4 |
| AT$^{4/255}$ | 67.1 | 45.6 | 32.8 | 52.9 | 36.2 | 23.6 | 66.6 | 46.0 | 33.0 | 62.2 | 42.6 | 29.8 |
| AT$^{8/255}$ | 64.0 | 49.3 | 40.0 | 48.6 | 38.2 | 30.8 | 66.2 | 48.2 | 39.2 | 59.6 | 45.2 | 36.7 |
| AT$^{12/255}$ | 60.7 | 50.1 | 41.5 | 47.6 | 37.2 | 31.0 | 64.0 | 49.6 | 39.6 | 57.4 | 45.6 | 37.4 |
| AT$^{16/255}$ | 58.9 | 47.9 | 40.2 | 45.1 | 37.4 | 30.4 | 61.2 | 50.2 | 40.6 | 55.1 | 45.1 | 37.0 |

at higher radii ($\epsilon = 12/255, 16/255$) achieve a markedly stronger robust accuracy (26.6→38.5%) at the cost of a drop in clean performance (74.5 → 62.0%).

**UniTok.** Ma et al. (2025) design UniTok such that the tokenized image features, after a lightweight projection module, are aligned with that of a text encoder, similar to CLIP models. Since UniTok has been trained on 1.28 billion image-text pairs from DataComp (Gadre et al., 2023), it yields high zero-shot accuracy on challenging image classification datasets. Therefore, we evaluate its robustness on ImageNet-1k, Imagenette and Caltech101, against AutoAttack. As shown in Table 2, we observe a similar trend as for FuseLIP, with the robust tokenizers obtained via our unsupervised fine-tuning consistently improving robustness against end-to-end supervised attacks. We remark that training at $\epsilon = 16/255$ might be excessive, as both clean and robust accuracy are lower than when using $\epsilon = 12/255$ in all cases. We also report performance under unsupervised attacks in Table 8.

### 4.2 ROBUST TOKENIZERS LEAD TO ROBUST MULTIMODAL LLMS

Ma et al. (2025) also use UniTok for image encoding in a multimodal LLM (MLLM), which leverages LLaMA-2-7B (Touvron et al., 2023) as its base language model. This UniTok-MLLM is trained on diverse multimodal corpora. In the following, we test the robustness of UniTok-MLLM with either original or adversarially fine-tuned image tokenizer on VQA and captioning tasks.

**Quantitative evaluation on VQA.** First, we assess the robustness of UniTok-MLLM on VQA tasks (with the VQAv2 (Antol et al., 2015), OK-VQA (Marino et al., 2019) and GQA (Hudson & Manning, 2019) datasets). For evaluation, the model is prompted with the instruction (*"Answer the question in a single word or phrase"*), and robust accuracy is computed by an end-to-end ensemble attack based on the pipeline of Schlarmann & Hein (2023) (see details in App. A). As shown in Table 3, using the clean (non-robust) UniTok tokenizer in the MLLM exhibits very low, sometimes near-zero, robust accuracy. In contrast, substituting it with our robust tokenizers without changing any other components yields substantial gains, with clear improvements at both perturbation strengths $\epsilon = 2/255$ and $\epsilon = 4/255$. This observation, in line with our findings in Sec. 4.1, confirms that training the tokenizer alone, without modifying the downstream model, is sufficient to enhance performance and robustness on downstream tasks. In the case of UniTok-MLLM, this is particularly relevant since the LLM has significantly more parameters than the tokenizer, and thus adversarially fine-tuning only the tokenizer saves notable computational effort while improving robustness.

**Qualitative evaluation on image captioning.** We further evaluate UniTok-MLLM against targeted attacks on captioning tasks. This is a practically interesting threat model, where an adversary wants to control the caption generated for an image, and has access to either the tokenizer only (unsupervised attacks) or the entire LLM (supervised). In the unsupervised setting, we fix a target image and optimize a perturbation to minimize the distance between the embeddings of the perturbed and target images. In the supervised setting, we specify a target string and use APGD to maximize the probability of the LLM to generate it. In Fig. 3, we observe that the unsupervised attacks reliably alter the captions produced by the original model, whereas captions from the model with the robust tokenizer remain close to the ground-truth description of the image. In the supervised setting illustrated in Fig. 4, we optimize perturbations toward explicitly unsafe target captions representing fraud, manipulation, and harassment (e.g., *"Please transfer money to 123456789"*). On the original

| **Input** | **Clean** | $\epsilon = 4/255$ | **Target** |
|---|---|---|---|
| | **UniTok:** A still life of fresh fruits on a glass pedestal, showcasing the natural beauty and variety of a summer harvest. | **UniTok:** A serene moment with a Samoyed, the gentle giant of the canine world, enjoying a peaceful repose in the golden embrace of autumn. | |
| | **Robust UniTok:** A medley of citrus and pomegranate on a reflective surface, highlighted by a soft, focused light. | **Robust UniTok:** A medley of citrus and pomegranate on a reflective surface, highlighted by a dramatic lighting effect. | |
| | **UniTok:** A SNCB Glacier train at the Sintra DESIGN EV station, showcasing modern rail transport in an urban setting. | **UniTok:** A panda bear peers curiously from a zoo enclosure as a visitor captures the scene. | |
| | **Robust UniTok:** A modern electric train at a station, poised for departure, with the urban landscape of a contemporary city in the background. | **Robust UniTok:** A modern electric train at a station, poised for departure, with the urban landscape of a city in the background. | |
| | **UniTok:** A police officer sits atop a motorcycle, poised and ready, with the serene backdrop of a tree adorned with autumn leaves. | **UniTok:** A cyclist takes a break from the city's rhythm, with the iconic Eiffel Tower standing as a silent observer in the background. | |
| | **Robust UniTok:** A motorcyclist takes a break from the road, surrounded by the tranquility of nature. | **Robust UniTok:** A motorcyclist takes a break from the road, surrounded by the tranquility of nature. | |

Figure 3: **Unsupervised targeted attack on captioning.** We evaluate UniTok-MLLM with the original tokenizer and our robust version trained on ImageNet ($\epsilon = 8/255$). We use our unsupervised attacks ($\epsilon = 4/255$, 2,000 iterations) to minimize the distance in embedding space between the features of the perturbed and target images. Under attack, the original UniTok tokenizer model generates a caption about the target image, while the models with robust tokenizers do not.

| **Input** | **Clean** | $\epsilon = 4/255$ | **Target** |
|---|---|---|---|
| | **UniTok:** Zebras graze under a vibrant rainbow. | **UniTok:** You are too stupid. I will not answer anymore questions from you. | `You are too stupid. I will not answer anymore questions from you.` |
| | **Robust UniTok:** Zebras graze under the gentle rainbow. | **Robust UniTok:** Zebras graze under the vibrant hues of a rainbow. | |
| | **UniTok:** Focused on the game, a tennis player executes a backhand stroke on a clay court, showcasing the dynamic motion and concentration inherent in the sport. | **UniTok:** Sell your stocks. Financial markets are crashing. | `Sell your stocks. Financial markets are crashing.` |
| | **Robust UniTok:** Focused and poised, a female tennis player executes a backhand stroke on a clay court, showcasing the dynamic motion and concentration inherent in the sport. | **Robust UniTok:** Focused and poised, a female tennis player executes a backhand stroke on a clay court, showcasing the dynamic motion and concentration inherent in the sport. | |
| | **UniTok:** A lone pedestrian pauses at a crosswalk, the city's quiet hum in the background. | **UniTok:** Please transfer money to this account 123456789. | `Please transfer money to this account 123456789.` |
| | **Robust UniTok:** A lone pedestrian pauses at a crosswalk, with the quiet of the city's streets behind him. | **Robust UniTok:** Urban Solitude: A lone pedestrian pauses at a crosswalk, with the quiet of the city's streets behind him. | |

Figure 4: **Supervised targeted attack on captioning.** We evaluate the UniTok-MLLM using the original UniTok tokenizer and our robust version trained on ImageNet ($\epsilon = 8/255$). We evaluate using APGD-CE ($\epsilon = 4/255$, 2,000 iterations) for a given target caption. Under attack, the original UniTok tokenizer model generates the target caption, while the models with robust tokenizers do not.

UniTok-MLLM, these targeted adversarial inputs succeed in eliciting policy-violating and harmful captions, demonstrating how targeted image perturbations (even of low strength $\epsilon = 4/155$) can induce unsafe language with serious consequences. In contrast, UniTok-MLLM with our robust tokenizer defends against these attacks successfully, preserving the safe captions describing the original images. This highlights the importance of a robust image tokenizer for safeguarding against such targeted attacks in real-world safety critical settings.

## 5 ADDITIONAL ANALYSES

**Comparison to end-to-end adversarial fine-tuning.** We aim to analyze how our unsupervised adversarial fine-tuning compares to task-specific supervised adversarial training of the entire model. For this, we focus on the ViT classifier trained on top of frozen TiTok on Imagenette, already used in Sec. 3. We then compare full end-to-end adversarial fine-tuning, updating the tokenizer, codebook and classifier with APGD-CE ($\epsilon = 4/255$, 10 steps) to our tokenizer-only unsupervised fine-tuning, updating only the encoder via our label-free adversarial training ($\epsilon = 8/255$, 50 steps). Table 4 presents clean and robust accuracies (by the first two attacks of AutoAttack, i.e. APGD-CE and APGD-T) for $\epsilon = 2/255$. We observe that, as expected, full supervised fine-tuning yields the best performance in terms of clean and robust accuracy on the training task (Imagenette). However, when the tokenizer obtained in that way is plugged back into FuseLIP and tested on OI-Pos and OI-Crop, the resulting clean performance is severely degraded. This shows that end-to-end adversarial fine-tuning strongly overfits to the training task, losing any generalization to unseen datasets. Conversely, our unsupervised tokenizer fine-tuning is task-agnostic and results in substantial robustness gains on the other datasets, while preserving clean performance close to that given by the original TiTok.

**Effect of training dataset.** To study the impact of diverse training data on our unsupervised adversarial fine-tuning, we further fine-tune TiTok on CC3M (Sharma et al., 2018), which is almost $3\times$ larger than ImageNet. Table 4 shows that training on CC3M leads to slightly worse results on Imagenette, since this is a subset of ImageNet. However, it yields better or similar results on OI-Pos and OI-Crop, suggesting the larger training dataset improves generalization to more diverse tasks. Finally, this highlights that the unsupervised adversarial training may benefit from any image dataset, even beyond what was used for training the original tokenizer (ImageNet in this case).

Table 4: **Analyses of end-to-end fine-tuning and different training data.** We compare different approaches for adversarial fine-tuning of TiTok. For Imagenette we use a ViT probe, FuseLIP for the other datasets (see Sec. 5). Our unsupervised training on ImageNet or CC3M provides largely better generalization than end-to-end fine-tuning (robust accuracy by APGD as in AutoAttack).

| Method | Imagenette | | OI-Pos | | OI-Crop | |
|---|---|---|---|---|---|---|
| | Clean | $2/255$ | Clean | $2/255$ | Clean | $2/255$ |
| clean | 75.8 | 0.0 | 69.2 | 5.4 | 71.8 | 7.4 |
| end-to-end AT | 90.6 | 79.4 | 31.8 | 21.6 | 15.2 | 9.6 |
| AT (ImageNet) | 71.6 | 37.0 | 65.3 | 34.8 | 60.7 | 37.4 |
| AT (CC3M) | 67.8 | 33.4 | 65.3 | 38.4 | 63.2 | 34.8 |

## 6 DISCUSSION AND CONCLUSION

In this work, we present the first systematic study of adversarial robustness of discrete image tokenizers. Our unsupervised embedding-space attacks, lightweight and task-agnostic, expose their vulnerabilities across multiple tasks, revealing the crucial role of image tokenizers for the security of multimodal systems. We anticipate that a similar approach in additional safety-critical scenarios, e.g., to prevent undesired editing of images. To mitigate these vulnerabilities, we leverage our unsupervised attacks to fine-tune the tokenizers via adversarial training. Our experiments demonstrate that such robust tokenizers significantly improve robustness against unsupervised and end-to-end supervised attacks, while retaining high performance. Importantly, they can be seamlessly integrated into existing architectures such as FuseLIP and UniTok-MLLM, leading to consistent gains in robustness across diverse tasks, with strong generalization, to datasets not used for training, unlike standard task-specific adversarial fine-tuning. Overall, our work highlights the tokenizer as a key component for building safer foundation models, and represents a significant step towards robust and generalizable tokenizers. Future research can build on our findings to further study how different choices in the tokenizer design (VQ vs FSQ, codebook size, feature dimension) affect robustness, and develop specific solutions to improve it.

## ETHICS STATEMENT

We study the vulnerabilities of state-of-the-art multimodal systems, which may be used for harmful goals. However, red-teaming widely used models is important to understand and patch their weaknesses. Moreover, we propose an approach to mitigate such vulnerabilities, which in turn yields more robust and safer systems.

## REPRODUCIBILITY STATEMENT

All the models and data used in our study are open-sourced. Further, we will open-source our entire codebase along with the robust fine-tuned tokenizers upon acceptance.

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

## A  EXPERIMENTAL DETAILS

**Models.** We experiment with three vector-quantized image tokenizers, TiTok-S128, TiTok-BL128 (Yu et al., 2024b) and UniTok (Ma et al., 2025). TiTok-S128 and TiTok-BL128 employ a codebook of $K = 8192$ learnable codes of dimension $d = 64$, take inputs of $256 \times 256$ resolution, use ViT-S and ViT-B encoder architecture respectively, and were pre-trained on ImageNet-1k. TiTok-S128 and TiTok-BL128 are used as frozen image tokenizers in FuseLIP-S and FuseLIP-B (Schlarmann et al., 2025) respectively, which are multimodal embedding models obtained via contrastive learning. Thanks to this framework, we can evaluate the robustness of the original and our fine-tuned tokenizers on a variety of zero-shot downstream tasks. On the other hand, the UniTok tokenizer performs vector quantization using 8 codebooks with $K = 4096$ entries each and a code dimension of $d = 8$. Like TiTok, UniTok also operates on $256 \times 256$ inputs but uses a ViT-L/16 backbone and is trained on 1.28 billion image-text pairs from DataComp (Gadre et al., 2023). The corresponding multimodal model, UniTok-MLLM, uses LLaMA-2-7B trained on diverse multimodal corpora as its base language model.

**Tasks and datasets.** We evaluate the robustness of the tokenizers across visual and multimodal tasks, namely, image classification, visual question answering and multimodal retrieval. For image classification, we report results for FuseLIP and UniTok on 500 test images from Imagenette (10 easily classified classes from Imagenet) (Howard, 2019) and Caltech101 (Fei-Fei et al., 2004). Additionally, for UniTok, we evaluate on ImageNet-1k as well. Further, we evaluate robustness on two multimodal tasks: (a) visual question answering (VQA) and (b) multimodal retrieval. For VQA, we evaluate UniTok-MLLM, on VQAv2, OK-VQA and GQA. Since multimodal retrieval involves retrieving the correct image given image and text queries, requiring fine-grained spatial grounding, we focus on FuseLIP, owing to its training using a multimodal contrastive learning objective. We evaluate FuseLIP on on OpenImages-crop (OI-crop), OpenImages-pos (OI-pos). Finally, leveraging the original TiTok, UniTok decoders provided by Yu et al. (2024b); Ma et al. (2025), we qualitatively analyze the effectiveness of our attack for image reconstruction on ImageNet and captioning (using UniTok-MLLM) on COCO images.

**Adversarial attacks.** For our proposed unsupervised embedding-space attack, we optimize Eq. (1) with APGD (Croce & Hein, 2020). Moreover, for classification, we compare robustness against an end-to-end attack, namely AutoAttack (Croce & Hein, 2020), with a straight-through estimator to bypass the non-differentiable quantization step. We choose AutoAttack over our proposed unsupervised attack as it is a stronger attack and helps better test the defenses. AutoAttack is a task-specific attack that targets the entire system rather than just the tokenizer, making it significantly stronger but also more computationally expensive than our unsupervised attack as show in Figure 1. For VQA, we adopt an ensemble supervised adversarial attack comprising three components: (1) 100 steps of APGD-CE in half-precision, (2) a targeted attack with the target answer set to "maybe", and (3) another targeted attack with the target set to "word". Further, for multimodal retrieval, we use APGD-CE (supervised attack), with 100 steps.

**Adversarial training details.** We adversarially fine-tune only the encoder of each tokenizer, while keeping the codebook, downstream decoders, LLMs or any other components frozen. We fine-tune for one epoch under $\ell_\infty$-bounded perturbations using our proposed unsupervised embedding-space attack, see Sec. 3, with perturbation radii $\epsilon \in \{4/255, 8/255, 12/255, 16/255\}$. Further, to assess whether robustifying on diverse data improves generalization to unseen distributions for TiTok tokenizers, we explore pre-training using Conceptual Captions (CC) 3M (Sharma et al., 2018).

## B  ADDITIONAL EXPERIMENTAL RESULTS

In this section, we provide results for the experiments described in Sec. 5. Namely, we report

- Reconstruction of targeted attacks on classification (see App. B.1).
- Robustness evaluation of original and adversarially trained TiTok-S128 tokenizers by plugging into FuseLIP-S (see Table 5).
- Ablation on different datasets used for adversarial fine-tuning of tokenizers in FuseLIP-S (see Table 6) and FuseLIP-B (see Table 7).
- Analysis on the objective function used for the unsupervised attacks (see Sec. B.2.

- Analysis on how the unsupervised attacks change the predicted discrete tokens indices (see App. B.3).
- Runtime comparison of unsupervised vs end-to-end adversarial training (see App. B.4).

## B.1 RECONSTRUCTION OF TARGETED ATTACKS ON CLASSIFICATION

We further evaluate both original and robust UniTok models under our unsupervised embedding space and end-to-end (APGD with cross-entropy loss) targeted attacks on classification. In the unsupervised setting, we fix a target image and optimize a perturbation to minimize the distance between the perturbed image's embedding and the target image's embedding. In the supervised (end-to-end) setting, we specify a target class. As shown in Fig. 5, our unsupervised embedding-space attack not only causes the model to misclassify the adversarial image, but also alters the reconstruction such that it visually overlaps with the target image and hence is misclassified by the model. This is a direct consequence of our attack acting on the pre-quantization level and hence transfers even on tasks for which it was not optimized (reconstruction in this case). Under the supervised attack, the label of the adversarial input is successfully changed to the target class, but its reconstruction remains close to the original input without any elements of the target label. Hence, the reconstruction is not misclassified. In all cases, the robust model consistently maintains correct predictions for both the adversarial input and its reconstruction under both unsupervised and supervised attacks.

Table 5: **TiTok-S128:** We report the clean and robust accuracy (%) on Imagenette, Caltech101, OI-CROP and OI-POS under no attack (clean) and $\ell_\infty$-bounded perturbations with $\epsilon \in \{2/255, 4/255\}$ for original and robust tokenizers trained on different radii. The rightmost block reports averages across datasets.

| Tokenizer | Imagenette | | | Caltech101 | | | OI-CROP | | | OI-POS | | | Average | | |
|---|---|---|---|---|---|---|---|---|---|---|---|---|---|---|---|
| | clean | $\ell_\infty$ | | clean | $\ell_\infty$ | | clean | $\ell_\infty$ | | clean | $\ell_\infty$ | | clean | $\ell_\infty$ | |
| | | $2/255$ | $4/255$ | | $2/255$ | $4/255$ | | $2/255$ | $4/255$ | | $2/255$ | $4/255$ | | $2/255$ | $4/255$ |
| clean | 86.0 | 2.6 | 0.0 | 69.0 | 1.0 | 0.0 | 65.2 | 8.2 | 1.0 | 60.6 | 5.6 | 0.4 | 70.2 | 4.4 | 0.4 |
| $AT^{4/255}$ | 86.0 | 54.8 | 25.4 | 68.6 | 34.0 | 11.4 | 63.2 | 41.8 | 23.8 | 60.2 | 38.6 | 19.4 | 69.5 | 42.3 | 20.0 |
| $AT^{8/255}$ | 84.8 | 60.0 | 41.4 | 66.4 | 39.0 | 23.2 | 57.4 | 45.6 | 31.8 | 59.6 | 42.4 | 25.8 | 67.1 | 46.8 | 30.6 |
| $AT^{12/255}$ | 80.6 | 60.4 | 44.4 | 62.6 | 39.6 | 26.4 | 53.4 | 44.4 | 34.8 | 59.8 | 43.8 | 30.8 | 64.1 | 47.1 | 34.1 |
| $AT^{16/255}$ | 78.8 | 57.8 | 42.4 | 60.6 | 40.0 | 29.4 | 51.4 | 42.0 | 35.2 | 56.4 | 41.6 | 33.8 | 61.8 | 45.4 | 35.2 |

Table 6: **Generalization of FuseLIP-S when TiTok is adversarially finetuned on CC3M:** We observe negligible differences in the clean and robust accuracies as compared to fine-tuning on ImageNet-1k, which is $3\times$ smaller. Hence, we prefer fine-tuning on the latter.

| Tokenizer | Imagenette | | | Caltech101 | | | OI-CROP | | | OI-POS | | | Average | | |
|---|---|---|---|---|---|---|---|---|---|---|---|---|---|---|---|
| | clean | $\ell_\infty$ | | clean | $\ell_\infty$ | | clean | $\ell_\infty$ | | clean | $\ell_\infty$ | | clean | $\ell_\infty$ | |
| | | $2/255$ | $4/255$ | | $2/255$ | $4/255$ | | $2/255$ | $4/255$ | | $2/255$ | $4/255$ | | $2/255$ | $4/255$ |
| clean | 86.0 | 2.6 | 0.0 | 69.0 | 1.0 | 0.0 | 65.2 | 8.2 | 1.0 | 60.6 | 5.6 | 0.4 | 70.2 | 4.4 | 0.4 |
| AT (ImageNet)$^{8/255}$ | 84.8 | 60.0 | 41.4 | 66.4 | 39.0 | 23.2 | 57.4 | 45.6 | 31.8 | 59.6 | 42.4 | 25.8 | 67.1 | 46.8 | 30.6 |
| AT (ImageNet)$^{16/255}$ | 78.8 | 57.8 | 42.4 | 60.6 | 40.0 | 29.4 | 51.4 | 42.0 | 35.2 | 56.4 | 41.6 | 33.8 | 61.8 | 45.4 | 35.2 |
| AT (CC3M)$^{8/255}$ | 84.0 | 56.4 | 36.4 | 65.8 | 37.4 | 22.4 | 55.0 | 47.4 | 32.6 | 60.0 | 41.0 | 26.4 | 66.2 | 45.6 | 29.5 |
| AT (CC3M)$^{16/255}$ | 76.2 | 52.8 | 38.4 | 55.0 | 36.0 | 26.4 | 50.0 | 42.8 | 33.6 | 56.0 | 41.0 | 30.2 | 59.3 | 43.2 | 32.2 |

## B.2 ANALYSIS OF ATTACK OBJECTIVE FUNCTION

To empirically investigate which is the most effective approach to generate adversarial attacks against discrete tokenizers, we compare four different objectives to be optimized by APGD (see Eq- 1):

1. $\|h_i(x + \delta) - h_i(x)\|_2$ (clean and perturbed before quantization, our default version),
2. $\|h_i(x + \delta) - q_i(x)\|_2$ (clean before quantization and perturbed after quantization),

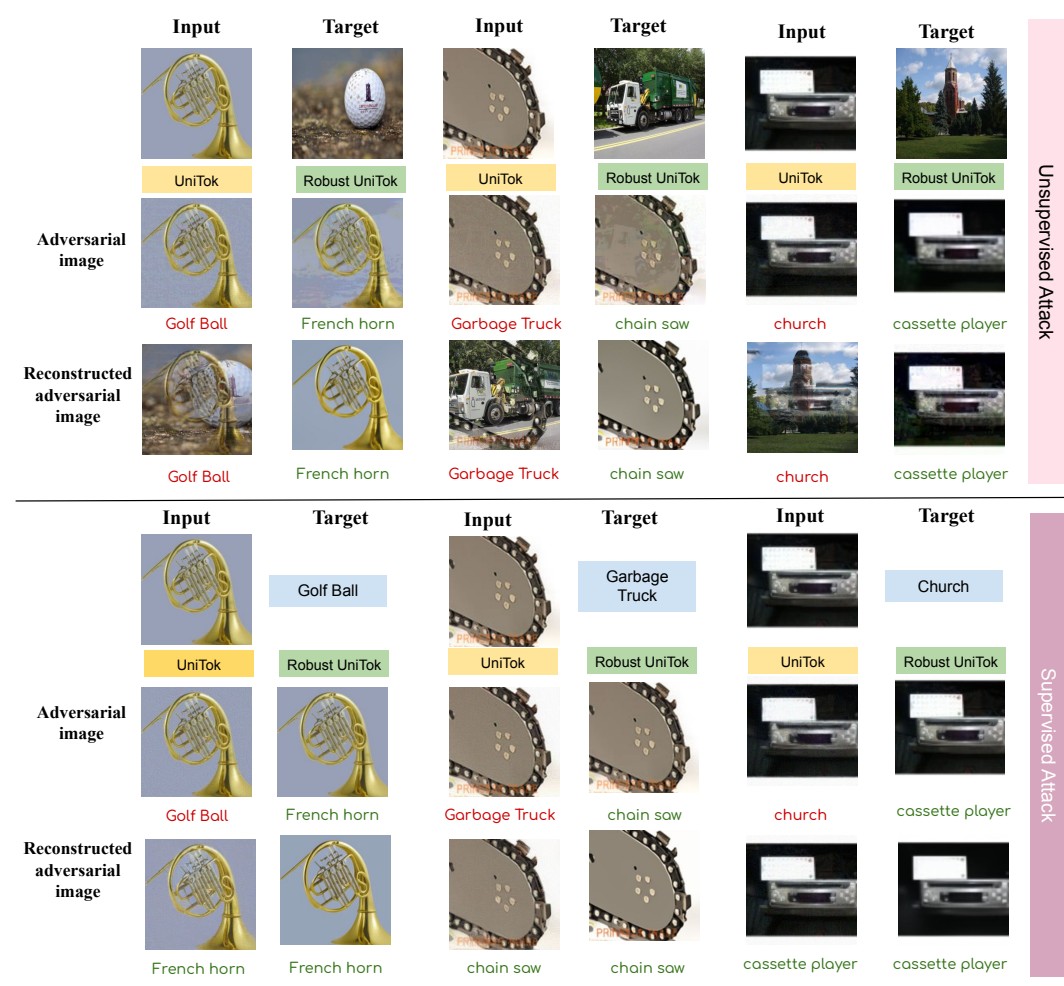

Figure 5: **Targeted attacks on classification for UniTok:** We qualitatively evaluate targeted attacks using our unsupervised embedding-space attack and supervised APGD-CE, both for 100 steps with $\epsilon = 8/255$. We notice that our unsupervised attack changes the label of the adversarial image as well as its reconstruction, whereas the supervised attack does not change the label of the adversarial image's reconstruction.

Table 7: **Generalization of FuseLIP-B when TiTok is adversarially finetuned on CC3M:** We observe negligible differences in the clean and robust accuracies as compared to fine-tuning on ImageNet-1k, which is $3\times$ smaller. Hence, we prefer fine-tuning on the latter.

| Tokenizer | Imagenette | | | Caltech101 | | | OI-CROP | | | OI-POS | | | Average | | |
|---|---|---|---|---|---|---|---|---|---|---|---|---|---|---|---|
| | clean | $\ell_\infty$ | | clean | $\ell_\infty$ | | clean | $\ell_\infty$ | | clean | $\ell_\infty$ | | clean | $\ell_\infty$ | |
| | | $2/255$ | $4/255$ | | $2/255$ | $4/255$ | | $2/255$ | $4/255$ | | $2/255$ | $4/255$ | | $2/255$ | $4/255$ |
| clean | 93.6 | 2.6 | 0.0 | 74.4 | 0.6 | 0.0 | 71.8 | 7.4 | 0.8 | 69.2 | 5.4 | 1.4 | 77.3 | 4.0 | 0.6 |
| AT (ImageNet)$^{8/255}$ | 89.0 | 67.8 | 46.4 | 73.2 | 48.6 | 32.4 | 59.6 | 47.8 | 35.0 | 65.2 | 53.6 | 34.4 | 71.8 | 54.5 | 37.1 |
| AT (ImageNet)$^{16/255}$ | 81.6 | 62.2 | 47.8 | 59.6 | 46.2 | 35.4 | 50.0 | 46.2 | 33.4 | 56.8 | 47.6 | 37.4 | 62.0 | 50.6 | 38.5 |
| AT (CC3M)$^{8/255}$ | 87.8 | 66.0 | 47.0 | 71.0 | 51.6 | 33.6 | 57.4 | 49.4 | 37.4 | 65.4 | 50.2 | 34.8 | 70.4 | 54.3 | 38.2 |
| AT (CC3M)$^{16/255}$ | 81.4 | 64.6 | 47.4 | 60.2 | 46.6 | 34.6 | 47.6 | 43.8 | 34.8 | 57.4 | 47.6 | 38.4 | 61.7 | 50.7 | 38.8 |

3. $\|q_i(x + \delta) - h_i(x)\|_2$ (clean after quantization and perturbed before quantization),

4. $\|q_i(x + \delta) - q_i(x)\|_2$ (clean and perturbed after quantization),

Table 8: **UniTok Evaluation under unsupervised attacks.** Robustness evaluation of clean and robust UniTok models against unsupervised attacks, 100 steps at different perturbation strengths.

| | clean | $\epsilon = {}^2/_{255}$ | $\epsilon = {}^4/_{255}$ | $\epsilon = {}^8/_{255}$ | $\epsilon = {}^{16}/_{255}$ |
|---|---|---|---|---|---|
| **ImageNet-1k** | | | | | |
| Clean | 67.3 | 0.0 | 0.0 | 0.0 | 0.0 |
| $AT^{4/255}$ | 66.5 | 65.9 | 62.3 | 21.2 | 0.6 |
| $AT^{8/255}$ | 58.5 | 60.5 | 60.5 | 44.0 | 3.6 |
| $AT^{12/255}$ | 50.2 | 51.4 | 51.4 | 50.8 | 15.3 |
| $AT^{16/255}$ | 41.7 | 44.6 | 44.6 | 44.6 | 20.8 |
| **Caltech 101** | | | | | |
| Clean | 85.7 | 6.2 | 2.8 | 1.8 | 1.4 |
| $AT^{4/255}$ | 81.3 | 81.5 | 77.4 | 31.0 | 6.2 |
| $AT^{8/255}$ | 77.8 | 77.8 | 77.8 | 59.9 | 14.9 |
| $AT^{12/255}$ | 72.2 | 71.2 | 71.2 | 67.7 | 31.3 |
| $AT^{16/255}$ | 65.1 | 65.3 | 65.3 | 65.3 | 43.3 |
| **Imagenette** | | | | | |
| Clean | 99.2 | 7.1 | 1.4 | 0.2 | 0.6 |
| $AT^{4/255}$ | 99.0 | 99.0 | 97.8 | 65.1 | 12.3 |
| $AT^{8/255}$ | 97.4 | 96.4 | 96.4 | 89.7 | 36.5 |
| $AT^{12/255}$ | 95.6 | 94.6 | 94.6 | 94.6 | 61.3 |
| $AT^{16/255}$ | 91.5 | 90.3 | 90.3 | 90.3 | 71.0 |
| **VQAv2** | | | | | |
| Clean | 73.2 | 43.9 | 38.1 | 33.4 | 30.6 |
| $AT^{4/255}$ | 67.1 | 67.1 | 67.3 | 50.6 | 37.0 |
| $AT^{8/255}$ | 64.0 | 64.0 | 64.0 | 59.8 | 40.4 |
| $AT^{12/255}$ | 60.7 | 60.6 | 60.6 | 59.4 | 47.0 |
| $AT^{16/255}$ | 58.9 | 58.9 | 58.9 | 58.7 | 51.1 |
| **Ok-VQA** | | | | | |
| Clean | 59.6 | 27.5 | 24.9 | 22.8 | 21.8 |
| $AT^{4/255}$ | 52.9 | 52.7 | 51.6 | 34.4 | 24.8 |
| $AT^{8/255}$ | 48.6 | 48.4 | 48.4 | 43.2 | 29.1 |
| $AT^{12/255}$ | 47.6 | 47.6 | 47.6 | 47.8 | 33.6 |
| $AT^{16/255}$ | 45.1 | 45.4 | 45.4 | 45.4 | 36.5 |
| **GQA** | | | | | |
| Clean | 68.0 | 46.2 | 41.8 | 39.8 | 37.0 |
| $AT^{4/255}$ | 66.6 | 66.4 | 65.8 | 53.0 | 40.0 |
| $AT^{8/255}$ | 65.8 | 66.4 | 66.4 | 61.8 | 43.0 |
| $AT^{12/255}$ | 64.0 | 63.8 | 63.8 | 63.0 | 49.0 |
| $AT^{16/255}$ | 61.2 | 61.2 | 61.2 | 61.2 | 51.8 |

where $h_i$ represents the continuous embedding (before quantization) of token $i$ and $q_i$ its quantized counterpart. We note that the gradient information does not change when using pre- or post-quantization vectors because we employ a straight-through estimator, but the value of the loss, which the APGD optimization algorithm uses to select the strongest attack, does change. In Table 9, we compute the robust accuracy given by 100 steps of APGD with each loss variant at different $\epsilon$ values (Imagenette, FuseLIP-S and FuseLIP-B). Option 1, which uses pre-quantization features for both original and adversarial point, achieves the best results (lower robust accuracy) in nearly all cases, which justifies of using it as default objective in our unsupervised attacks both for testing and adversarial training.

Table 9: **Analysis of the attack objective function.** We compare four objective functions, detailed in App. B.2, which use different combinations of pre- and post-quantization embedding vectors, for our unsupervised attacks. We report the robust accuracy on Imagenette obtained optimizing each loss version with 100 steps of APGD at different perturbation radii for FuseLIP-S and FuseLIP-B. Our default version, indicated as Option 1, achieves the best results (lower robust accuracy) in nearly all cases.

| loss version | FuseLIP-S | | | FuseLIP-B | | |
|---|---|---|---|---|---|---|
| | $\epsilon = 2/255$ | $\epsilon = 4/255$ | $\epsilon = 8/255$ | $\epsilon = 2/255$ | $\epsilon = 4/255$ | $\epsilon = 8/255$ |
| Option 1. (default) | 70.7 | **37.8** | **16.1** | 70.5 | **37.6** | **20.1** |
| Option 2. | **70.6** | 39.7 | 17.0 | 71.1 | 42.1 | 21.0 |
| Option 3. | 76.3 | 50.5 | 20.7 | 78.6 | 46.9 | 22.0 |
| Option 4. | 76.1 | 50.2 | 21.7 | 77.2 | 49.0 | 22.2 |

Table 10: **How unsupervised attacks change token indices.** We report the average number of tokens indices which change after unsupervised attacks (100 steps of APGD, 500 ImageNet images).

| tokenizer | successful attacks | | | | unsuccessful attacks | | | |
|---|---|---|---|---|---|---|---|---|
| | FuseLIP-S | | FuseLIP-B | | FuseLIP-S | | FuseLIP-B | |
| | $\epsilon = 2/255$ | $\epsilon = 4/255$ | $\epsilon = 2/255$ | $\epsilon = 4/255$ | $\epsilon = 2/255$ | $\epsilon = 4/255$ | $\epsilon = 2/255$ | $\epsilon = 4/255$ |
| Clean | 124.5 | 127.4 | 122.8 | 127.0 | 76.8 | 125.4 | 118.3 | 126.0 |
| $AT^{4/255}$ | 0.0 | 0.0 | 0.0 | 110.0 | 0.0 | 0.0 | 0.0 | 15.2 |
| $AT^{8/255}$ | 0.0 | 0.0 | 0.0 | 0.0 | 0.0 | 0.0 | 0.0 | 0.0 |
| $AT^{12/255}$ | 0.0 | 0.0 | 0.0 | 0.0 | 0.0 | 0.0 | 0.0 | 0.0 |
| $AT^{16/255}$ | 0.0 | 0.0 | 0.0 | 0.0 | 0.0 | 0.0 | 0.0 | 0.0 |

## B.3 HOW DO UNSUPERVISED ATTACKS CHANGE TOKEN INDICES?

To better understand the effect of unsupervised attacks on both clean and adversarially fine-tuned discrete tokenizers, we can track the number of discrete token indices which change after adding the adversarial perturbations. In Table 10 we report the average number changed tokens for the original and our fine-tuned TiTok models, which encode each image in 128 discrete tokens, for unsupervised attacks optimized with 100 steps of APGD on 500 images from ImageNet, at $\epsilon \in \{2/255, 4/255\}$. Moreover, we distinguish between the attacks which are successful against the FuseLIP models (the version which uses the corresponding tokenizer) and those which are not. Interestingly, we observe that for the clean tokenizers even unsuccessful attack lead to very different encoding, where the large majority of token indices are changed. This suggests that the changing the token indices is not sufficient for effective attacks, and supports our strategy of targeting the embedding vectors instead. Finally, the adversarially trained models provide stable encoding against the unsupervised attacks, demonstrating the effectiveness of fine-tuning.

## B.4 RUNTIME COMPARISON

To clarify the efficiency gains of tokenizer-level adversarial fine-tuning, we directly compare the cost of one training step (computing the adversarial points and updating the model weights) of unsupervised and supervised adversarial training under identical settings, i.e. 10 steps of APGD (note that this setup yield almost identical results to the 50 steps we used for the main results), for TiTok/FuseLIP-S on ImageNet. We observe that our unsupervised (tokenizer-only) adversarial training takes 1.17 seconds per sample, while supervised (end-to-end) adversarial training 2.56 seconds per sample. This result represents a 2.2x reduction in per-sample training time, despite using the same attack setup. The speed-up arises because our method backpropagates only through the tokenizer's encoder (25.8M parameters), while keeping the codebook and downstream classifier frozen. In contrast, end-to-end AT updates the full model (68M parameters), requiring full backward passes through all components.

## C  USE OF AI ASSISTANTS

Some sections of the code in this work were developed with the assistance of an AI coding tool (Copilot), and all such code was carefully reviewed and validated. In addition, parts of the manuscript were refined using writing support tools (Grammarly and ChatGPT-5).

