# OpenReview forum: "On the Adversarial Robustness of Discrete Image Tokenizers"
_ICLR.cc/2026/Conference — Submitted to ICLR 2026_

### Official Review · Reviewer_dhLJ · 2025-10-28

**Soundness:** 2
**Presentation:** 2
**Contribution:** 2
**Rating:** 2
**Confidence:** 4

**Summary:**

This paper investigates the adversarial robustness of discrete image tokenizers, an area noted as underexplored. The authors first propose an unsupervised adversarial attack targeting these tokenizers. Second, based on this attack, the authors propose an unsupervised adversarial fine-tuning strategy to enhance tokenizer robustness.

**Strengths:**

- This is the first work to systematically study the adversarial robustness of discrete image tokenizers.
- The experimental results demonstrate the effectiveness of the proposed unsupervised adversarial fine-tuning.
- The defense method only requires fine-tuning the tokenizer's encoder, while keeping the much larger downstream components like LLMs frozen.

**Weaknesses:**

- As shown in Figure 1, the unsupervised attack performs comparably to, or even weaker than, standard end-to-end supervised attacks, especially at small $\epsilon$ values. The authors do not demonstrate that this attack uncovers new vulnerabilities that supervised attacks miss. I didn't see the motivation. Is the only reason unsupervison?
The authors claim the attack is 'unsupervised' and 'task-agnostic'. However, this property is not unique to tokenizer models. Unsupervised attacks on the embedding space of other VLMs like CLIP are also feasible. It is therefore unclear why this 'unsupervised' property is particularly relevant to tokenizers.

- The proposed unsupervised adversarial fine-tuning strategy also appears to lack methodological novelty. The core idea is to minimize the distance between the embeddings of perturbed samples and the embeddings of the original samples (from the original tokenizer) . This is a well-established paradigm in robust representation learning, bearing a strong resemblance to the core idea of works like TRADES [1]. The proposed loss in this paper is almost identical to the robustness part of TRADES. The paper seems to apply a known framework to a new model (tokenizers) without sufficiently discussing its connections and differences from prior work.

[1] Zhang, Hongyang, et al. "Theoretically principled trade-off between robustness and accuracy." International conference on machine learning. PMLR, 2019.

- The paper's core focus is on discrete image tokenizers. However, both the proposed attack and defense operate in the continuous embedding space ($h_i$) before quantization. While the paper mentions the quantization step and its non-differentiable nature, it does not actually exploit or target this discretization process with a unique attack or defense strategy. The current methods seem applicable to any encoder-based VLM (like CLIP) without modification, which weakens the paper's specific contribution to "discrete tokenizers."

**Questions:**

- Given the results in Figure 1, can the authors further clarify why it was necessary to propose a new unsupervised attack (Eq. 1)  instead of just using known, stronger supervised attacks for adversarial training? If the goal was simply to enable unsupervised training, why not adapt existing unsupervised attacks on embedding spaces (like those for CLIP )?

- Can the authors elaborate on the differences between their training objective and TRADES [1], or other unsupervised/self-supervised AT methods? What is the core methodological innovation of this work?

- Did the authors attempt to directly attack the discrete token indices ($q_i$), perhaps by using Gumbel-Softmax or other gradient estimation techniques to bypass the non-differentiability? Would an attack on the discrete space expose different vulnerabilities than the continuous embedding space attack?

---

> ### Author Response · Authors · 2025-11-25
> **Response to Reviewer dhLJ (1/3)**
>
> We thank the reviewer for the detailed feedback. We are glad that the reviewer appreciated the novel problem definition, the extensive empirical evaluation, and the efficiency of our defense. In the following, we address the individual questions raised in the review.
>
>
> > “As shown in Figure 1, the unsupervised attack performs comparably to, or even weaker than, standard end-to-end supervised attacks… The authors do not demonstrate that this attack uncovers new vulnerabilities that supervised attacks miss. I didn't see the motivation. Is the only reason unsupervison? [...] Unsupervised attacks on the embedding space of other VLMs like CLIP are also feasible. It is therefore unclear why this 'unsupervised' property is particularly relevant to tokenizers. ”
>
> We believe there is a misunderstanding about the message of Fig. 1. Its goal is not to show that the proposed unsupervised attack uncovers vulnerabilities that supervised attacks miss, but rather that it can achieve similar similar results to supervised attacks while (i) using no task-specific information (e.g., labels) and (ii) acting only on the image tokenizer instead of the end-to-end model. This observation is crucial for our main contribution: it justifies using unsupervised attacks for task-agnostic adversarial training. If such unsupervised attacks were not effective, adversarially training the tokenizer with them would not yield downstream robustness to supervised attacks. Our results show that this is not the case, and that unsupervised adversarial training is an efficient way to improve the robustness of models that integrate discrete image tokenizers.
>
> We agree that unsupervised attacks are not unique to discrete image tokenizers. However, we focus on these tokenizers because they are increasingly used as shared components in larger systems (multimodal LLMs, FuseLIP, etc.), and thus improving their robustness can therefore benefit many downstream models at once. Therefore, testing and improving the adversarial robustness of discrete image tokenizers is a relevant and timely topic, so far unexplored.
>
> Our work also reveals a phenomenon that is specific to discrete image tokenizers: the vulnerability of the reconstruction, which has no direct counterpart in models such as CLIP. We can leverage the image decoder to visualize the reconstructed adversarial points, see Fig. 2. Additionally, Fig. 5 in Appendix shows that our unsupervised targeted attacks change both the classifier’s label of the adversarial image and the label of its reconstruction (with a visually drastic shift toward the target), whereas supervised attacks using the standard classification loss do not change the label of the adversarial image's reconstruction.
>
> > “The proposed loss in this paper is almost identical to the robustness part of TRADES…”
>
>
> We respectfully disagree with the statement that our loss is “almost identical to the robustness part of TRADES”. TRADES is designed to control the robustness-accuracy trade-off of image classifiers by jointly optimizing a supervised classification loss (cross-entropy) on clean images and a robustness regularizer that penalizes KL divergence between the classifier’s softmax outputs on clean and adversarial samples.
>
> Our setting differs from TRADES in several fundamental ways. First, the TRADES framework cannot translate to our unsupervised adversarial fine-tuning setup with tokenizers because we do not have a supervised component (no labels are used by tokenizers). Second, the robustness part of TRADES is defined between two probability distributions (softmax outputs), which we do not have in image tokenizers. In fact, image tokenizers output discrete codes obtained by nearest-neighbor lookup in a codebook with respect to the L2 distance. Consequently, to improve the robustness of image tokenizers, we minimize the L2-distance between the continuous embedding of clean and adversarial inputs. Third, our goal is to obtain robust tokenizers that can be plugged into multiple downstream models without further fine-tuning, whereas TRADES is inherently task- and model-specific, tailoring robustness to a particular classifier.
>
> Altogether, our approach represents a fundamental algorithmic shift compared to TRADES, with different goals and techniques.

---

> > ### Author Response · Authors · 2025-11-25
> > **Response to Reviewer dhLJ (2/3)**
> >
> > > “Can the authors elaborate on the differences between their training objective and TRADES [1], or other unsupervised/self-supervised AT methods? What is the core methodological innovation of this work?”
> >
> > In Sec. 3.2 we acknowledge that our unsupervised adversarial loss is inspired by the one used by Schlarmann et al. (2024) for CLIP encoders. However, in our case, the L2-loss is justified by the fact that in the discretization step, the continuous embedding vectors are mapped to the discrete codebook via L2-projection. Therefore, if the L2-distance between the embedding of clean and adversarial images is sufficiently small, then the predicted discrete tokens will be identical, and so will be the output of any downstream model built on top of the discrete tokenizer. Notably, this differs from the case of continuous embedding, where the difference between the embedding of clean and adversarial images, unless it is exactly zero, will propagate to the output of downstream models.
> >
> >
> > > “the proposed attack and defense operate in the continuous embedding space before quantization. While the paper mentions the quantization step and its non-differentiable nature, it does not actually exploit or target this discretization process…”
> >
> > While our attack and defense operate in the continuous embedding space before quantization, we argue that this choice is deliberate, as it yields the strongest attacks against discrete tokenizers, as we show in the following analysis. To empirically validate our approach, we compare attacks optimized by APGD with four different objectives:
> >
> > - $||h_i(x + \delta) - h_i(x)||_2$ (clean and perturbed before quantization, **our default version**)
> > - $||h_i(x + \delta) - q_i(x)||_2$ (clean before quantization and perturbed after quantization)
> > - $||q_i(x + \delta) - h_i(x)||_2$ (clean after quantization and perturbed before quantization)
> > - $||q_i(x + \delta) - q_i(x)||_2$ (clean and perturbed after quantization)
> >
> > where $h_i$ represents the continuous embedding (before quantization) of token $i$ and $q_i$ its quantized counterpart. We note that the gradient information does not change when using pre- or post-quantization vectors because we employ a straight-through estimator, but the value of the loss, which the APGD optimization algorithm uses to adapt the step size and select the strongest attack, does change.
> >
> > Below, we report the robust accuracy of original FuseLIP-S (Table A) and FuseLIP-B (Table B) models under each attack variant for various $\epsilon$ values. Across both models and all epsilons, Option 1, the one we used in the experiments in the paper for both evaluation and adversarial training, consistently gives the strongest attack, i.e., lowest robust accuracy. This shows that perturbing and comparing features before quantization is the most damaging configuration, and therefore the most appropriate point to attack and defend in discrete tokenizers.
> >
> > **Table A.** Robust accuracy of FuseLIP-S against unsupervised APGD attacks with different objective functions.
> >
> > | $\epsilon$       | **Option 1** | Option 2 | Option 3 | Option 4 |
> > |---------|----------|----------|----------|----------|
> > | 2/255   | 70.7    | **70.6**    | 76.3    | 76.1    |
> > | 4/255   | **37.8**    | 39.7    | 50.5    | 50.2    |
> > | 8/255   | **16.1**    | 17.0    | 20.7    | 21.7    |
> >
> > **Table B.** Robust accuracy of FuseLIP-B against unsupervised APGD attacks with different objective functions.
> >
> > | $\epsilon$       | Option 1 | Option 2 | Option 3 | Option 4 |
> > |---------|----------|----------|----------|----------|
> > | 2/255   | **70.5**     | 72.1     | 78.6     | 77.2     |
> > | 4/255   | **37.6**     | 42.1     | 46.9     | 49.0     |
> > | 8/255   | **20.1**     | 21.0     | 22.0     | 22.2     |

---

> > > ### Author Response · Authors · 2025-11-25
> > > **Response to Reviewer dhLJ (3/3)**
> > >
> > > > “Did the authors attempt to directly attack the discrete token indices, perhaps by using Gumbel-Softmax or other gradient estimation techniques to bypass the non-differentiability? Would an attack on the discrete space expose different vulnerabilities than the continuous embedding space attack?”
> > >
> > > To gauge the feasibility of attacks targeting the discrete token indices, we track how many indices are changed, on average, by our unsupervised attacks against TiTok models (we use 100 steps of APGD, 500 ImageNet validation points). Moreover, we distinguish between successful (Table C) and unsuccessful (Table D) attacks against FuseLIP models that use TiTok models as tokenizers. Most notably, the results in Table D show that, for clean models, even unsuccessful attacks change almost all the token indices (TiTok encodes each image in 128 tokens). Therefore, changing (even all) the token indices is not sufficient to obtain misclassification, since there is no control over how different the resulting post-quantization features are. Thus, attacks directly targeting the discrete token indices, e.g., via Gumbel-Softmax, would not fully exploit the vulnerabilities of the image tokenizers. These results (also added to App. B.3 in the revised manuscript), together with the analysis above, support our strategy of optimizing the distance between the pre-quantization features to obtain the strongest attacks. Nevertheless, exploring more sophisticated loss functions for unsupervised attacks may be an interesting direction for future work.
> > >
> > > **Table C.** Average change in tokens after an unsupervised attack is *successful* on 500 images of ImageNet-1k.
> > >
> > > | Tokenizer     | attack $\epsilon$ | clean     | AT $\epsilon$=4/255 | AT $\epsilon$=8/255 | AT $\epsilon$=12/255 | AT $\epsilon$=16/255 |
> > > |-----------|----------|-----------|------------------------|------------------------|------------------------|----------------|
> > > | TiTok-S128 | 2/255    | 124.5 | 0              | 0              | 0               | 0               |
> > > | TiTok-S128 | 4/255    | 127.5 | 0              | 0              | 0               | 0               |
> > > | TiTok-BL128 | 2/255    | 122.8 | 0               | 0              | 0               | 0               |
> > > | TiTok-BL128 | 4/255    | 127.0 | 110.0          | 0              | 0               | 0               |
> > >
> > > **Table D.** Average change in tokens after an unsupervised attack is *unsuccessful* on 500 images of ImageNet-1k.
> > >
> > > | Tokenizer     | attack $\epsilon$ | clean     | AT $\epsilon$=4/255 | AT $\epsilon$=8/255 | AT $\epsilon$=12/255 | AT $\epsilon$=16/255 |
> > > |-----------|----------|-----------|------------------------|---------------------------|------------------------|--------------|
> > > | TiTok-S128 | 2/255    | 76.8  | 0              | 0              | 0               | 0               |
> > > | TiTok-S128 | 4/255    | 125.4 | 0              | 0              | 0               | 0               |
> > > | TiTok-BL128 | 2/255    | 118.3 | 0              | 0              | 0               | 0               |
> > > | TiTok-BL128 | 4/255    | 126.0 | 15.19   | 0              | 0               | 0               |
> > >
> > >
> > > &nbsp;
> > >
> > > We thank the reviewer again and hope that we have addressed their concerns. We also hope that the original score can be reconsidered in light of our additional analysis and clarifications.

---

> > ### Comment · Reviewer_dhLJ · 2025-11-26
> >
> > Could the authors further clarify why the proposed method is fundamentally different from TRADES‐style approaches? The paper states that “our method minimizes the L2 distance between the continuous embeddings of clean and adversarial inputs” and that “the TRADES framework cannot be directly applied to our unsupervised adversarial fine-tuning setup with a tokenizer, because there is no supervised component (the tokenizer does not use any labels)”.
> >
> > However, the robustness regularization term in TRADES is the KL divergence between the softmax outputs on clean and adversarial samples, which does not require labels.
> >
> > In other words, do the authors agree that the proposed objective can be viewed as a TRADES-like framework without the cross-entropy loss on clean samples, and with the KL distance replaced by an Euclidean (L2) distance? If not, could the authors provide further explanation? I am still confused about this point.

---

> ### Author Response · Authors · 2025-11-26
> **Response to Reviewer dhLJ**
>
> We thank the reviewer for the quick response.
>
> The original TRADES objective can be written as
>
> $$
> \min_f  E_{(x,y)}[ \ell_{CE}(f(x),y) ]
> +
> \beta E_{x}[ \max_{\||\delta\||\le \epsilon} \mathrm{KL}( f(x) \|| f(x+\delta) ) ]
> $$
>
>
> where the core goal is to explicitly trade off natural (clean) accuracy and adversarial robustness of a classifier $f$. In this sense, TRADES is fundamentally a supervised method: both its theoretical analysis and practical behavior rely on the presence of the clean cross-entropy term.
>
> We agree with the reviewer that the KL regularization term in TRADES is itself label-free. In this broad sense, our loss and the TRADES regularizer are both instances of adversarial consistency regularization. However, simply removing the supervised term from TRADES and replacing the KL with an $\ell_2$ distance yields an objective with a very different role and interpretation from the original TRADES framework.
>
> There is another important difference in our setting compared to TRADES: in Eq. (2) of our paper, the clean example is always encoded by the *original tokenizer (frozen)* $h^{original}$, while the adversarial example is encoded by the *robust tokenizer (fine-tuned)* $h_\theta$ (with a simplified notation which omits the sum over tokens):
>
> $$
> L =
> \min_\theta E_x[
> \max_{\||\delta\||\le \epsilon}
> ||  h_{\theta}(x+\delta) - h^{original}(x) ||_{2}^2
> ]
> $$
>
>
>
> The loss therefore enforces that the robust tokenizer $h_{\theta}$ matches the behavior of the original tokenizer $h^{original}$ on clean inputs, even under adversarial perturbations. This constraint is essential to our application: it guarantees that the robust tokenizer remains compatible with downstream systems (multimodal LLMs, FuseLIP) that were pretrained with $h^{original}$, without any further alignment.
>
> In contrast, TRADES compares a single model to itself, $\mathrm{KL}(f(x)\|\|f(x+\delta))$, and completely relies on the supervised cross-entropy term to maintain high accuracy on clean samples. The classifier is then used in isolation, not as a drop-in replacement for a component in a larger pretrained system.
>
> For these reasons, we would describe our objective as *related to* the unsupervised regularization term used in TRADES, but not as "TRADES without the cross-entropy term and with KL replaced by $\ell_2$". The main conceptual difference is that we (i) work in a fully unsupervised setting and (ii) try to explicitly match a robust tokenizer to the original tokenizer so that it can be plugged into larger systems without retraining them.
>
> We will revise the paper to clarify this relationship more carefully and to explicitly discuss similarities and differences to TRADES in the related work section.

---

### Official Review · Reviewer_v5pr · 2025-11-03

**Soundness:** 3
**Presentation:** 3
**Contribution:** 3
**Rating:** 6
**Confidence:** 1

**Summary:**

The paper systematically investigates the adversarial robustness of discrete image tokenizers used in multimodal systems. It introduces an unsupervised embedding-space attack that maximizes the distance between original and perturbed images in the pre-quantization feature space, effectively altering codebook indices without relying on labels or downstream supervision. Building on this attack objective, the authors further perform adversarial fine-tuning by updating only the tokenizer encoder while keeping downstream components frozen, thereby enhancing robustness under ℓ∞-bounded perturbations for models such as FuseLIP and UniTok-MLLM. Comprehensive experiments demonstrate the accuracy–robustness trade-off and cross-task transferability across classification, retrieval, and vision–language generation tasks. The results demonstrates that unsupervised tokenizer-level robustness training can generalize across diverse downstream tasks, achieving both efficiency and task-agnostic applicability, in contrast to traditional end-to-end adversarial training.

**Strengths:**

- **Problem Motivation and Scope:**
  The paper is well-motivated, addressing a critical yet underexplored vulnerability in multimodal foundation models. The focus on discrete image tokenizers—now ubiquitous in modern vision and vision-language pipelines—is both timely and highly relevant.

- **General, Task-Agnostic Defense:**
  The work introduces an unsupervised adversarial fine-tuning strategy that operates entirely at the tokenizer level and requires only unlabeled images. The resulting robust tokenizers can be seamlessly integrated into architectures such as FuseLIP and UniTok-MLLM, providing broad robustness improvements. Extensive quantitative evaluations cover classification, retrieval, and VQA tasks, while qualitative analyses demonstrate improved resistance to targeted captioning attacks.

- **Empirical Rigor and Diversity:**
  The paper provides strong empirical support, including multi-dataset evaluations, ablation studies, and visualizations of adversarial effects. The qualitative analyses (e.g., Figures 3 and 4) compellingly illustrate both targeted and untargeted safety vulnerabilities—such as policy breaches in captioning—and the corresponding effectiveness of the proposed defense.

- **Clarity and Writing Quality:**
  The paper is clearly written and well-structured. The motivation and problem formulation are easy to follow, and the experimental results are presented concisely with well-designed figures and tables highlighting key findings. The background and references are appropriate, making the work accessible even to readers less familiar with adversarial robustness research.

**Weaknesses:**

1. **Minor Grammatical and Typographical Errors:**
   - “captioninig” → “captioning” (Section 4.2, “VQA and captioninig tasks”)
   - “severaly degraded” → “severely degraded” (Table 4 discussion, “the resulting clean performance is several[y] degraded”)
   - “unsupervsied” → “unsupervised” (Discussion section, “improve robustness against unsupervsied and end-to-end supervised attacks”)
   - “imputs” → “inputs” (Related work section, “extend masked modeling losses to visual imputs”)

2. **No Explicit Runtime or Efficiency Analysis:**
   The rationale for attacking/fine-tuning at the tokenizer level is partly computational, suggesting notable savings compared to full model adversarial training. However, the paper provides no quantitative runtime or resource comparison, leaving readers uncertain about the true efficiency benefits.

**Questions:**

**Questions for the Authors:**

1. **Efficiency Comparison:**
   Since one of the motivations for tokenizer-level adversarial fine-tuning is computational efficiency, could the authors provide quantitative evidence—such as training time, GPU hours, or memory usage—comparing this approach with full end-to-end adversarial training? This would clarify the practical magnitude of the claimed efficiency gains.

2. **Comparison to Other Discrete Adversarial Methods:**
   Have the authors run, or could they run for the final version, a controlled comparison with other discrete adversarial or robustness-enhancing methods—such as **Discrete Adversarial Training (DAT)** or **manifold regularization** adapted to modern tokenizers? What trade-offs (e.g., computational cost, clean accuracy degradation, robustness generalization) would the authors expect in such a comparison?

---

> ### Author Response · Authors · 2025-11-25
> **Response to Reviewer v5pr**
>
> We thank the reviewer for their positive feedback and constructive suggestions. We are glad that the reviewer appreciated the well-motivated and novel problem, the defense design, the extensive empirical evaluation, and the quality of the presentation. In the following, we address the individual questions raised in the review.
>
> > “Efficiency Comparison: … could the authors provide quantitative evidence—such as training time, GPU hours, or memory usage—comparing this approach with full end-to-end adversarial training?”
>
> We thank the reviewer for this helpful suggestion. To clarify the efficiency gains of tokenizer-level adversarial fine-tuning, we directly compare the cost of one training step (computing the adversarial points and updating the model weights) of unsupervised and supervised adversarial training (AT) under identical settings, i.e., 10 steps of APGD (note that this setup yields almost identical results to the 50 steps we used for the results in the manuscript), for TiTok/FuseLIP-S on ImageNet. We observe that unsupervised (tokenizer-only) AT (ours) takes 1.17 seconds per sample, while supervised (end-to-end) AT takes  2.56 seconds per sample. This result represents a **2.2x** reduction in per-sample training time, despite using the same attack setup. The speedup arises because our method backpropagates only through the tokenizer’s encoder (25.8M parameters), while keeping the codebook and downstream classifier frozen. In contrast, end-to-end AT updates the full model (68M parameters), requiring full backward passes through all components. We have added this discussion to the App. B.4 in the revised manuscript.
>
> Beyond training efficiency, our approach offers key practical advantages over supervised fine-tuning:
> - **Task-agnostic:** It does not require labels or a specific training objective, making it applicable across classification, retrieval, and captioning tasks without modification. This also allows us to use any source of images to make the tokenizer robust. We demonstrated this by training on CC3M and Imagenet-1k dataset, as shown in Table 4.
> - **Better generalization:** As shown in Table 4, tokenizer-level AT consistently generalizes better across downstream tasks and datasets compared to full supervised AT, which often overfits to the training dataset.
> - **Modular and scalable:** It improves robustness in a plug-and-play fashion. Once the tokenizer is fine-tuned, it can be reused across multiple downstream models without retraining.
>
> > “Comparison to Other Discrete Adversarial Methods”
>
> First, we note that DAT [1] trains task-specific models on top of the output of a VQGAN, that is, the original image is encoded, quantized, and finally decoded before being passed to the model. In this framework, only the downstream model is trained, while the weights of the image tokenizer (encoder) are not updated, and thus the tokenizer does not become robust. This is in stark contrast to our approach, where we train only the discrete tokenizer to be adversarially robust, without involving any downstream model. Then, thanks to the unsupervised loss, the adversarially trained tokenizer can be plugged into existing systems and improve the robustness of these systems on downstream tasks. This use case is the main focus of our work, and cannot be handled by DAT, since it does not update the tokenizer. Therefore, DAT has different goals than our method (it can be seen as closer to supervised AT), and thus, a direct comparison is not possible.
>
> About manifold regularization, we are not aware of any standard method specifically adapted to discrete image tokenizers. If the reviewer could point to specific methods that could be of interest, we would be happy to comment on them.
>
> [1] Mao, X. et al. "[Enhance the visual representation via discrete adversarial training](https://arxiv.org/abs/2209.07735)". Advances in Neural Information Processing Systems, 2022
>
>
> > “Minor Grammatical and Typographical Errors”
>
> Thanks for letting us know. We have fixed these in the revised version.
>
>
> &nbsp;
>
> We thank the reviewer again. We hope that we have addressed their concerns and that they will consider improving their scores. Otherwise, we hope they can provide us with guidance on how to better address their concerns.

---

### Official Review · Reviewer_vyU7 · 2025-11-03

**Soundness:** 2
**Presentation:** 3
**Contribution:** 3
**Rating:** 6
**Confidence:** 3

**Summary:**

This paper is the first to study the adversarial robustness of discrete image tokenizers used in multimodal foundation models.
It proposes unsupervised embedding-space attacks that are efficient and task-agnostic, and uses them for unsupervised adversarial fine-tuning to improve tokenizer robustness.
Experiments on classification, retrieval, VQA, and captioning show large robustness gains without harming clean accuracy, with good generalization to unseen tasks and clear safety benefits.

**Strengths:**

- Novel problem definition: First dedicated work on adversarial robustness for discrete image tokenizers, an important but overlooked component in multimodal systems.
- Well-motivated and efficient method: Attack design is simple, label-free, and computationally less expensive compared to end-to-end attacks.
- Strong empirical coverage: Extensive experiments across diverse downstream tasks and datasets (classification, retrieval, VQA, captioning) validate both attacks and defenses.

**Weaknesses:**

- The attack and defense methods employed are effective but relatively classical (APGD, standard adversarial training)
- The study currently focuses on a small set of tokenizer architectures (TiTok, UniTok). Exploring a wider variety of tokenizer designs — such as different quantization schemes (VQ vs FSQ), codebook sizes, number of tokens, or hybrid architectures — would help assess whether the proposed defense generalizes across structural variations and reveal design factors influencing robustness.

**Minor Comments**
- Figures could explicitly state attack parameters for clarity.

**Questions:**

1. Could the proposed robustness improvements hold against newer or structurally different attacks beyond ℓp-norm bounded perturbations?
2. Would your tokenizer-level robustness improvement generalize to structurally different tokenizers, including varied quantization techniques and token lengths?

**Details Of Ethics Concerns:**

The study responsibly discloses vulnerabilities and provides mitigations, following red-teaming principles. Demonstrated misuse scenarios (e.g., targeted caption generation with malicious content) emphasize importance of defenses. The release of robust tokenizers with documentation poses minimal risk and aligns with responsible ML safety practices.

---

> ### Author Response · Authors · 2025-11-25
> **Response to Reviewer vyU7 (1/2)**
>
> We thank the reviewer for their positive feedback and constructive suggestions. We are glad that the reviewer appreciated the novel problem definition, the well-motivated and efficient attack design, and the extensive empirical evaluation. In the following, we address the individual questions raised in the review.
>
>
> > “The attack and defense methods employed are effective but relatively classical (APGD, standard adversarial training)”
>
> We acknowledge that our method builds on established algorithms. Our contribution is to demonstrate how they can be successfully applied in a previously unexplored yet practically relevant setting: discrete image tokenizers (both as standalone models and as part of more complex architectures). For this, we customize the loss functions for both unsupervised attacks and adversarial training (see the analysis of pre- vs post-quantization embeddings in the loss added in App. B.2). Moreover, we uncover phenomena specific to image tokenizers, such as the effect of unsupervised vs supervised attacks on reconstruction (Fig. 5 in App. B.1) and the effect of unsupervised attacks on predicted token indices (added in App. B.3).
>
> > “The study currently focuses on a small set of tokenizer architectures (TiTok, UniTok). Exploring a wider variety of tokenizer designs — such as different quantization schemes (VQ vs FSQ), codebook sizes, number of tokens, or hybrid architectures — would help assess whether the proposed defense generalizes across structural variations and reveal design factors influencing robustness.”
>
> First, we would like to emphasize that the three tokenizers we test (TiTok-S, TiTok-B, UniTok)  already cover different codebook sizes (In particular, UniTok uses multiple codebooks, unlike standard VQ strategies), number and structure of tokens (1D vs 2D), token dimension, and encoder architecture, as summarized in **Table A** below. Thus, we believe our setup already shows that our defense approach generalizes across structural variations. Moreover, we focus on these tokenizers because they are integrated into other models (FuseLIP, UniTok-MLLM). This integration allows us to study how robustness generalizes to downstream tasks.
>
> Following the reviewer's suggestion, we further extend our evaluation with a preliminary study, where we apply our proposed defense to [FlexTok](https://arxiv.org/abs/2502.13967). Since no classifier is available for FlexTok, we train a linear probe on top of the clean tokenizer on the Imagenette dataset, then we fine-tune the tokenizer with unsupervised adversarial training (200 steps of APGD, $\epsilon=8/255$), and use the same linear classifier. When evaluated with AutoAttack ($\epsilon=2/255$), using the original encoder yields 3.4% (96.8% clean performance), while the robust tokenizer gives 41.0% (83.0% clean). Thus, our approach also benefits FlexTok (e.g., this uses FSQ instead of VQ), and we believe that training on the entire ImageNet-1k will reduce the gap in clean performance. We are happy to expand these experiments and include them in the final version.
>
> **Table A.** Summary of the characteristics of the tokenizers used in our work.
>
> | Tokenizer     | Codebook Size        | Number of Tokens | Token Dimension | Encoder Architecture            | Tokenization Type | Quantization Technique |
> |-----------|-----------------------|------------------|-----------------|---------------------------------|--------------------|-------------------------|
> | TiTok-S   | 4096   | 128    | 12    | ViT-small                       | 1D                 | VQ                      |
> | TiTok-B   | 8192    | 128    | 64   | ViT-base                        | 1D                 | VQ                      |
> | UniTok    | 32768 (4096 x 8)      | 256 x 8    | 64     | ViTamine-large | 2D                 | VQ   |
> | FlexTok   | 64000  | 1–256 | 6    | VAE enc. + 18 transformer blocks | 1D          | FSQ  |

---

> > ### Author Response · Authors · 2025-11-25
> > **Response to Reviewer vyU7 (2/2)**
> >
> > > “Could the proposed robustness improvements hold against newer or structurally different attacks beyond $\ell_p$-norm bounded perturbations?”
> >
> > Previous work [A] has shown that adversarial training with respect to $\ell_\infty$-bounded perturbations improves the robustness of image classifiers against a variety of attacks, such as adversarial patches, frames, fog, snow, etc. Therefore, we expect that robust tokenizers would also achieve similar gains. To confirm this, we perform a preliminary experiment evaluating three non-$\ell_p$-bounded attacks, i.e., snow, patch, and frame attacks, on FuseLIP-S. As shown in **Table B** below, using our robust tokenizer in the pre-trained FuseLIP model yields consistently better robust accuracy than the clean tokenizer across attack types.
> >
> > **Table B.** Our tokenizer fine-tuned with unsupervised adversarial training wrt $\ell_\infty$ ($\epsilon=8/255$) yields higher robustness also against non $\ell_p$-bounded attacks.
> > | Model              | Clean | $\ell_\infty$ | Snow | Patch | Frame |
> > |--------------------|:-----:|:-----:|:----:|:-----:|:-----:|
> > | Clean FuseLIP-S    | **86.0** | 2.6 | 62.8 | 44.6 | 19.8 |
> > | Robust FuseLIP-S   | 84.2 | **67.8** | **80.4** | **60.8** | **45.8** |
> >
> > [A] Croce, F., Hein M. "[Adversarial Robustness against Multiple and Single $ l_p $-Threat Models via Quick Fine-Tuning of Robust Classifiers](https://arxiv.org/abs/2105.12508)". International Conference on Machine Learning, 2022
> >
> >
> > > “Figures could explicitly state attack parameters for clarity.”
> >
> > Thanks for the suggestion, we have added this in the revision.
> >
> > &nbsp;
> >
> > We thank the reviewer again. We hope that we have addressed their concerns and that they will consider improving their scores. Otherwise, we hope they can provide us with guidance on how to better address their concerns.

---

### Author Response · Authors · 2025-11-25
**General comment to all reviewers**

We thank all reviewers for their thoughtful evaluations and constructive feedback. We are encouraged that the reviewers found our **problem definition novel** (vyU7, v5pr, dhLJ), **well-motivated** (vyU7, v5pr), and appreciated the **efficiency and effectiveness** of our proposed methods (vyU7, v5pr, dhLJ). We are also grateful for the positive remarks on our **extensive empirical evaluation** (vyU7, v5pr, dhLJ) and the **clarity of presentation** (v5pr).

In response to the reviewers' suggestions, we have conducted several new experiments and analyses, including:
- Evaluation of additional tokenizer architectures and quantization schemes.
- Detailed analysis of the objective function of the unsupervised attacks.
- Investigation of the effect of unsupervised attacks on predicted discrete token indices.
- Quantitative runtime comparison between tokenizer-level and end-to-end adversarial training.

We have also uploaded a revised manuscript with updates marked in color for ease of reference. We hope that these additions adequately address the raised concerns, and we would be happy to clarify any remaining questions.

---

### Meta-Review · Area_Chair_yfAL · 2025-12-28

**Summary:**

This paper studies adversarial robustness of discrete image tokenizers via unsupervised embedding-space attacks and tokenizer-level adversarial fine-tuning. It was reviewed by three reviewers. While the reviewers agreed the problem is timely and the empirical evaluation is extensive, one reviewer raised high-confidence concerns about limited methodological novelty, arguing that the approach closely mirrors existing adversarial training or consistency-based methods applied to a new component. The other two reviewers viewed the results favorably but reported low to moderate confidence. These differing confidence levels, together with unresolved concerns about novelty and tokenizer-specific contribution, informed the final decision.

**Reviewer Concerns:**

The rebuttal addressed several concerns related to implementation and evaluation, including clarifications on the motivation for unsupervised attacks, the choice of pre-quantization objectives, additional experiments on diverse tokenizer architectures and non $\ell_p$-bounded attacks, and a more explicit discussion of computational efficiency.

However, the core concern raised by the most critical reviewer regarding limited methodological novelty and the close resemblance to existing adversarial training or consistency-regularization frameworks remains outstanding. In particular, questions about whether the approach fundamentally exploits the discrete nature of tokenizers, rather than applying established techniques in a new setting, were not fully resolved.

**Reviewer Scores:**

Among the three reviewers, two provided scores of 6 (marginally above the acceptance threshold), with confidence levels of 3 and 1 respectively. Given the rebuttal, these reviewers might have maintained their original scores, as their main concerns were largely addressed, but their relatively low confidence suggests they were unlikely to strongly advocate for acceptance.

The third reviewer issued a clear reject (score 2) with high confidence (4) and strong reservations about novelty and conceptual contribution, and would likely have maintained this score even with full participation in the discussion, as these core concerns were not fully resolved.

---

### Decision · Program_Chairs · 2026-01-26

Reject